

# Construction and validation of a 15-gene ferroptosis signature in lung adenocarcinoma

Guangxu Tu[1,2], Weilin Peng[1,2], Qidong Cai[1,2], Zhenyu Zhao[1,2], Xiong Peng[1,2], Boxue He[1,2], Pengfei Zhang[1,2], Shuai Shi[1,2], Yongguang Tao[3,4] and Xiang Wang[1,2]

[1] Department of Thoracic Surgery, The Second Xiangya Hospital, Central South University, Changsha, Hunan, China
[2] Hunan Key Laboratory of Early Diagnosis and Precise Treatment of Lung Cancer, Department of Thoracic Surgery, The Second Xiangya Hospital, Central South University, Changsha, Hunan, China
[3] Key Laboratory of Carcinogenesis and Cancer Invasion, Ministry of Education, Department of Pathology, Xiangya Hospital, Central South University, Changsha, Hunan, China
[4] NHC Key Laboratory of Carcinogenesis (Central South University), Cancer Research Institute and School of Basic Medicine, Central South University, Changsha, Hunan, China

## ABSTRACT

**Background:** Ferroptosis is a novel form of programmed cell death characterized by the excessive accumulation of intracellular iron and an increase in reactive oxygen species. Emerging studies have shown that ferroptosis plays a vital role in the progression of lung adenocarcinoma, but the effect of ferroptosis-related genes on prognosis has been poorly studied. The purpose of this study was to explore the prognostic value of ferroptosis-related genes.

**Methods:** Lung adenocarcinoma samples were downloaded from The Cancer Genome Atlas (TCGA) and Gene Expression Omnibus (GEO) databases. The least absolute shrinkage and selection operator (LASSO) Cox regression algorithm was used to establish a predictive signature for risk stratification. Kaplan–Meier (K–M) survival analysis and receiver operating characteristic (ROC) curve analysis were conducted to evaluate the signature. We further explored the potential correlation between the risk score model and tumor immune status.

**Results:** A 15-gene ferroptosis signature was constructed to classify patients into different risk groups. The overall survival (OS) of patients in the high-risk group was significantly shorter than that of patients in the low-risk group. The signature could predict OS independent of other risk factors. Single-sample gene set enrichment analysis (ssGSEA) identified the difference in immune status between the two groups. Patients in the high-risk group had stronger immune suppression, especially in the antigen presentation process.

**Conclusions:** The 15-gene ferroptosis signature identified in this study could be a potential biomarker for prognosis prediction in lung adenocarcinoma. Targeting ferroptosis might be a promising therapeutic alternative for lung adenocarcinoma.

Corresponding author
Xiang Wang, wangxiang@csu.edu.cn

## INTRODUCTION

Lung cancer has the highest morbidity and mortality worldwide (*Bray et al., 2018*). The most common subtype is lung adenocarcinoma (LUAD), accounting for approximately 40% of all lung cancers (*Wei et al., 2018*). Due to the lack of obvious clinical symptoms, most patients are diagnosed at relatively advanced stages and have 5-year survival rates of less than 15% (*Denisenko, Budkevich & Zhivotovsky, 2018*). In recent years, targeted therapies, such as epidermal growth factor tyrosine kinase inhibitors (EGFR-TKIs), have achieved much success (*Li et al., 2016*). However, drug resistance, which is an inevitable problem, causes the prognosis of LUAD patients to be far from satisfactory, and the survival rate at 5 years is still only 21% (*Macheleidt et al., 2018*). Therefore, it is meaningful to explore novel mechanisms of therapy and identify an effective prognostic model for risk stratification to improve the clinical outcomes of LUAD patients.

Iron is an indispensable element for human biological processes, while iron metabolism plays a dual role in the proliferation and death of tumor cells (*Wang et al., 2019b*). An increased level of iron within a limited range facilitates the proliferation of cancer cells, while the excessive accumulation of iron leads to the death of cancer cells by lipid peroxidation of the cell membrane, namely, ferroptosis (*Mou et al., 2019*; *Stockwell et al., 2017*). Ferroptosis is a novel form of regulated cell death and has a tumor suppressive function (*Dixon et al., 2012*). Emerging studies, though limited, have shown that ferroptosis plays a pivotal role in the regulation of tumor progression in non-small cell lung cancer (NSCLC). For example, SLC7A11, which could negatively regulate the process of ferroptosis, was reported to be overexpressed in LUAD and closely associated with tumor progression (*Hu et al., 2020*; *Ji et al., 2018*; *Ma et al., 2021*). Upregulated GPX4 was reported to promote the proliferation of cancer cells and play a role in the resistance to ferroptosis in NSCLC (*Ji et al., 2018*). High expression of NFS1 in LUAD could protect cancer cells from ferroptosis (*Alvarez et al., 2017*). High expression of FSP1, EGLN1 and STRYK1 was found to be associated with greater ferroptosis resistance in lung cancer cells (*Doll et al., 2019*; *Jiang et al., 2017*; *Lai et al., 2019*). In recent years, ferroptosis was reported to interact with some immune cells to influence tumor progression. For example, abnormal ferroptosis-mediated cell death could induce neutrophil recruitment and the inflammatory response to cancer cell death (*Pentimalli et al., 2019*). Additionally, Wang et al. revealed that CD8+ T cells drive the ferroptosis of cancer cells and enhance the antitumor effect, indicating a vital role of ferroptosis in human anticancer immunity (*Wang et al., 2019a*; *Zhang et al., 2020*). All these discoveries shed light on ferroptosis as a promising target for cancer therapy. However, there are still limited studies exploring the potential role of ferroptosis-related genes in LUAD so far, and their effect on prognosis remains largely unknown.

In this study, we comprehensively analyzed the expression patterns, prognostic value, biological functions and potential pathways of ferroptosis-related genes to gain a better understanding of ferroptosis in LUAD. Then, we constructed a prognostic signature of the ferroptosis-related genes in The Cancer Genome Atlas (TCGA) cohort and validated it in

two Gene Expression Omnibus (GEO) datasets. Finally, we further explored the correlation between the prognostic signature and the immune status of LUAD patients. Our findings may provide useful information for further studies in this field.

## METHODS

### Acquisition of ferroptosis-related genes

We downloaded the list of ferroptosis pathway genes (map04216) from the Kyoto Encyclopedia of Genes and Genomes (KEGG) pathway database (https://www.genome.jp/kegg/pathway.html) and designated these genes as ferroptosis-related genes. Iron metabolism-associated genes were retrieved from the R-HAS-917937 pathway in the Reactome pathway database (https://reactome.org/) and the cellular iron ion homeostasis pathway in the AmiGO2 database (http://amigo.geneontology.org/amigo). After systematically searching and analyzing the original document, we discarded the genes that do not have a modulatory effect on ferroptosis. In addition, we collected and integrated newly reported ferroptosis-related genes for subsequent research.

### Data collection

We obtained the level 3 mRNA expression profiles and corresponding clinical data of LUAD patients from the TCGA database (https://portal.gdc.cancer.gov/) and GEO database (https://www.ncbi.nlm.nih.gov/geo/) (*Rousseaux et al., 2013*; *Schabath et al., 2016*) up to September 20, 2020. Samples with a follow-up time of less than 30 days or lack of prognostic data were excluded. Since the TCGA and GEO databases are public to researchers and we completely abided by the publication guidelines as well as the policies of access to the database, ethical review and approval were not required.

### Identification of differentially expressed ferroptosis genes

The "limma" R package was used for the normalization of gene expression matrixes. Then, we matched the mRNA sequencing data with the ferroptosis-related gene list and performed differential expression analysis between the tumor tissues and normal tissues in the TCGA cohort by using false discovery rate (FDR) < 0.05 as the threshold. Thus, ferroptosis-related differentially expressed genes (DEGs) were identified. A heatmap and volcano plot to visualize the DEGs were generated by the "pheatmap" R package. In addition, the protein-protein interaction (PPI) network of the ferroptosis-related DEGs was analyzed in the Search Tool for the Retrieval of Interacting Genes (STRING) online database and visualized in Cytoscape 3.8.2 software. We also generated a correlation network of the DEGs using the "igraph" R package.

### Establishment and validation of the prognostic model

DEGs with prognostic value were screened by univariate Cox regression analysis with a *P* value less than 0.05. Then, the least absolute shrinkage and selection operator (LASSO) Cox regression algorithm was conducted to establish the signature. In brief, we used the normalized expression data as the independent variable and the overall survival (OS) data of the patients in the TCGA cohort as the response variable to perform the LASSO

algorithm for the shrinkage of variables by the "glmnet" R package. Tenfold cross-validation was used to narrow the number of candidate genes and identify the penalty parameter ($\lambda$), which corresponds to the lowest position of the likelihood deviance curve. We further evaluated the prognostic value of the LASSO genes by Kaplan–Meier (K–M) survival analysis. The K–M survival analysis was conducted using the optimal cutoff determined by the "surv_cutpoint" function in the "survival" R package.

The risk score of each patient was calculated with the following formula: risk score = $\Sigma$ (ExpmRNAn $\times$ $\beta$mRNAn). The patients were stratified by the median risk score into high- and low-risk groups. After that, we performed principal component analysis (PCA) to evaluate the discriminatory ability by the "status" R package. We evaluated the predictive ability of the signature for OS through K–M survival analysis as well as time-dependent receiver operating characteristic (ROC) curve analysis conducted by the "survminer" and "survivalROC" R packages. Finally, we performed multivariate Cox regression analysis to identify independent risk factors for OS. In this study, the TCGA cohort was used as the derivation cohort, and the GSE72094 and GSE30219 cohorts were used as external validation cohorts.

## Construction and evaluation of the predictive nomogram

We constructed a nomogram to predict the survival probabilities at 1, 2 and 3 years in the TCGA cohort by integrating all the independent risk factors using the "rms" R package. The patients were stratified into different risk groups, and K–M survival analysis was conducted to analyze the OS difference between the different risk groups. Then, we calculated the C index and performed time-dependent ROC analysis to further validate the prediction accuracy of the nomogram. Moreover, a calibration plot was used to evaluate the consistency between the predicted survival probability and the real observation. The GSE72094 cohort was used for the external validation of the nomogram.

## Correlation with immune status

Using the "GSVA" R package, we calculated the enrichment score of 16 immune-related cells as well as 12 immune-related functions for each patient by single-sample gene set enrichment analysis (ssGSEA). In brief, using a set of genes that correspond to a particular immune cell or immune function, the enrichment score was calculated in the gene expression matrix through the ssGSEA algorithm, and the enrichment scores were normalized for subsequent analysis. Then, the enrichment scores for diverse immune cells and functions of patients in different risk groups were compared to illustrate the potential correlation between ferroptosis and immune status. The R script used for this part of the analysis is provided in the GitHub website (https://github.com/guangxu0109/ssGSEA.git).

## Functional enrichment analysis

To gain insight into the molecular mechanisms of these ferroptosis-related genes, we performed Gene Ontology (GO) and KEGG enrichment analyses of the DEGs between the high-risk group and low-risk group in the TCGA cohort, which were screened by the

thresholds of |log2 fold change (FC)| ≥ 1 and FDR < 0.05. GO enrichment analysis was performed by the "clusterProfiler" R package, and KEGG enrichment analysis was performed by gene set enrichment analysis (GSEA) using GSEA 4.0.2 software. The pathways with a $P$ value less than 0.05 were considered to be significantly enriched.

### Statistical analysis

All statistical analyses were conducted in R software version 4.0.2. The Mann–Whitney test was conducted to compare gene expression levels between tumor tissues and normal tissues as well as to compare the ssGSEA enrichment scores between different risk groups. A $P$ value less than 0.05 (if not otherwise specified) was considered statistically significant.

## RESULTS

### Twenty-two genes correlated with prognosis were identified in the TCGA cohort

We generated a flowchart to describe the design of this study, as shown in Fig. 1. A total of 948 LUAD patients from the TCGA database ($N = 477$), GSE72094 cohort ($N = 386$) and GSE30219 cohort ($N = 85$) were used for subsequent analysis in this study. Table S1 shows the basic characteristics of these patients. A total of 125 ferroptosis-associated genes (as listed in Table S2) were identified to intersect with the mRNA expression matrix of the TCGA and GEO databases. Ninety-seven genes were confirmed to be differentially expressed between tumor tissues and nontumorous tissues (62 upregulated and 35 downregulated) in the TCGA cohort (Figs. 2A, 2B and Table S3). The PPI network and the correlation network of the DEGs are shown in Fig. S1. Twenty-two genes were identified to be correlated with prognosis through univariate Cox regression analysis (Fig. 3).

### Establishment and assessment of the prognostic ferroptosis signature in the TCGA cohort

A signature consisting of 15 genes was established based on the minimum λ value identified by the LASSO algorithm in the TCGA cohort (Fig. S2). The risk score of each patient was calculated with the following formula: ($-0.037 \times$ expression level of AGER) + ($0.261 \times$ expression level of CISD1) + ($-0.019 \times$ expression level of DPP4) + ($0.173 \times$ expression level of EGLN1) + ($0.077 \times$ expression level of FANCD2) + ($-0.253 \times$ expression level of GLS2) + ($-0.233 \times$ expression level of ISCU) + ($0.087 \times$ expression level of ITGA6) + ($0.024 \times$ expression level of ITGB4) + ($0.118 \times$ expression level of KRAS) + ($0.109 \times$ expression level of NEDD4) + ($-0.048 \times$ expression level of PEBP1) + ($-0.133 \times$ expression level of SLC11A2) + ($0.013 \times$ expression level of TFAP2A) + ($0.208 \times$ expression level of VDAC1). We further performed K–M survival analysis for the 15 ferroptosis-related genes in the prognostic signature. The results confirmed that all these genes were significantly correlated with OS (Fig. S3). Among them, AGER, DPP4, GLS2, ISCU, PEBP1 and SLC11A2 were identified to be protective factors for OS, while the remaining factors (CISD1, EGLN1, FANCD2, ITGA6, ITGB4, KRAS, NEDD4,

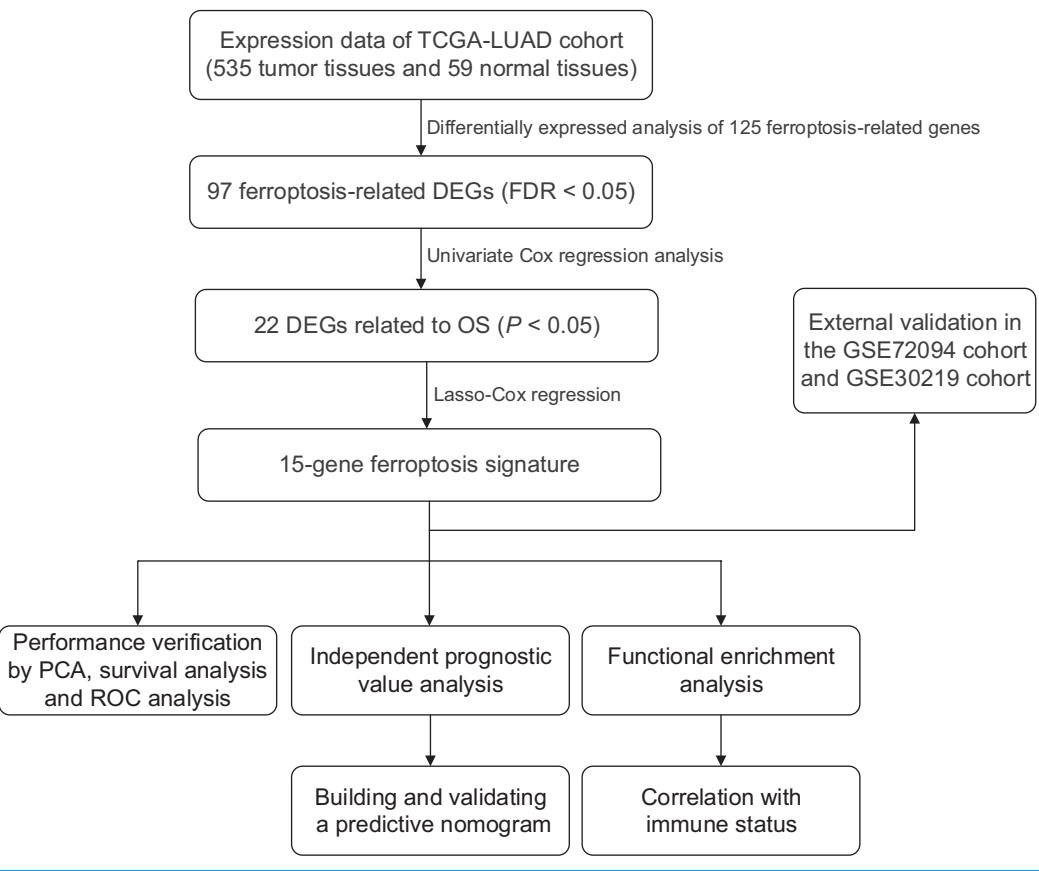

**Figure 1 Work flow of the study.** The TCGA cohort was used to construct the prognostic ferroptosis-related gene signature. The GSE72094 cohort and GSE30219 cohort were used to further validate the prognostic signature. TCGA, The Cancer Genome Atlas; LUAD, lung adenocarcinoma; DEGs, differentially expressed genes; FDR, false discovery rate; OS, overall survival; PCA, principal component analysis; ROC, the receiver operating characteristic.

TFAP2A and VDAC1) were risk factors, which was consistent with the results of univariate Cox analysis, as shown in Fig. 3.

The patients in the TCGA cohort were divided into different risk groups by the median risk score (Table S4). The PCA plot in Fig. 4A shows that patients in different risk groups were clearly distributed in two directions, indicating that the gene signature had good discriminatory power. The K–M curve showed that the OS of patients in the high-risk group was significantly lower than that of patients in the low-risk group (Figs. 4B, 4C). In addition, OS was significantly decreased with increasing risk scores (Fig. 4D). The area under the curve (AUC) value of the risk score model to predict OS was 0.775, which was the highest among all the risk factors, indicating the superior predictive ability of the prognostic signature (Fig. 5E).

## External validation of the ferroptosis-related gene signature

To further validate the robustness of the signature, we applied the 15-gene ferroptosis signature to the GSE72094 and GSE30219 cohorts. The patients in the two cohorts were stratified into high- and low-risk groups by the median risk score (Tables S5 and S6).

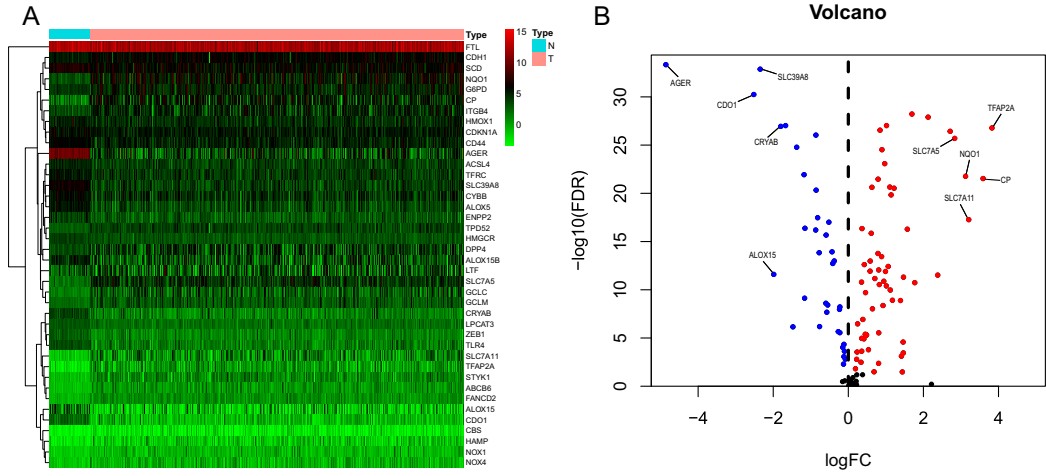

**Figure 2 Heatmap and volcano plot of the ferroptosis-related DEGs.** (A) Heatmap of the top 20 upregulated and downregulated ferroptosis-related genes between normal and tumor tissues in the TCGA-LUAD cohort. (B) Volcano plot of the 97 ferroptosis-related DEGs in the TCGA-LUAD cohort. The top five up- and down-regulated ferroptosis-related DEGs were labeled in the volcano plot. FDR<0.05 was set as screening criteria. The blue, red and black dots represented the down-, up-regulated ferroptosis- related DEGs and genes that were not satisfied the screening criteria, respectively. DEGs, differentially expressed genes; FDR, false discovery rate; TCGA, The Cancer Genome Atlas; LUAD, lung adenocarcinoma; N, normal tissues; T, tumor tissues.

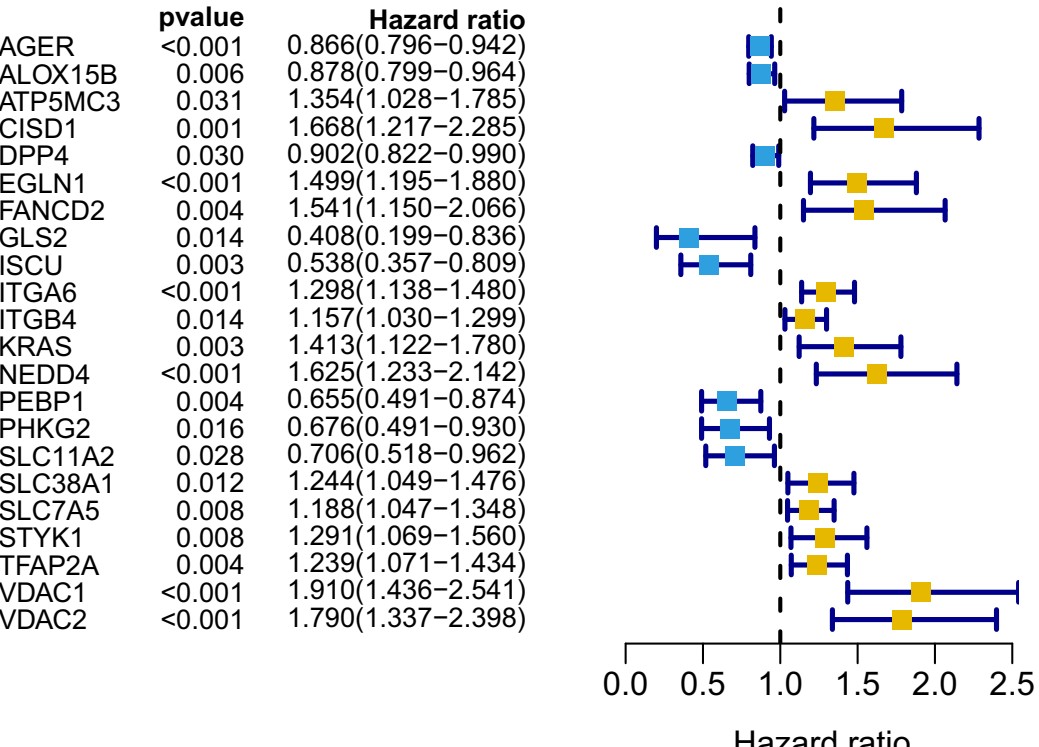

**Figure 3 Forest map of 22 prognostic ferroptosis-related DEGs screened by univariate Cox regression.** DEGs, differentially expressed genes. Yellow: risk factors; blue: protective factors.

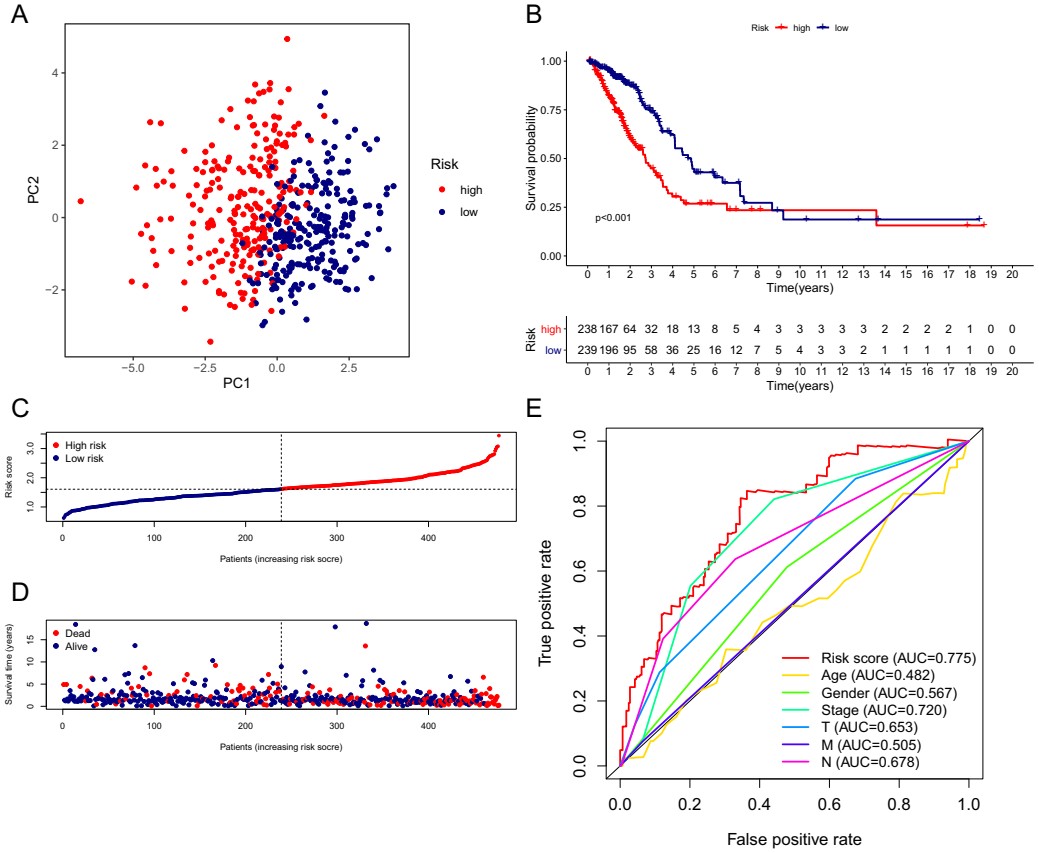

**Figure 4 Performance verification of the 15-gene ferroptosis signature in the TCGA cohort.** (A) PCA plot of patients in high-risk group (red) and low-risk group (navy blue). (B) Kaplan–Meier curve of patients in high-risk group (red) and low-risk group (navy blue). (C) The distribution of risk score of each patient. (D) The distribution of OS, survival status and risk score of each patient. Patients' OS was decreased with the increase of risk score. (E) Time-dependent ROC curves of risk score and clinical characteristics. TCGA, The Cancer Genome Atlas; PCA, principal component analysis; OS, overall survival; AUC, area under curve; ROC, receiver operating characteristic.

In the GSE72094 cohort, the PCA plot showed that patients in different risk groups were distributed in two directions (Fig. 5A). The K–M curve indicated that patients in the high-risk group had significantly shorter OS than those in the low-risk group, and OS was significantly decreased with increasing risk scores (Figs. 5B–5D). The AUC value of the risk score model was 0.711, which was the highest among all the risk factors (Fig. 5E). In the GSE30219 cohort, the PCA plots showed that patients in different groups were distributed in two directions (Fig. 5F). The K–M survival curve confirmed a significantly worse OS for patients in the high-risk group than for those in the low-risk group (Fig. 5G). OS significantly decreased with increasing risk scores (Figs. 5H, 5I). The AUC value of the risk score model was 0.877, which was the highest among all the risk factors (Fig. 5J). All these results indicated that the 15-gene ferroptosis signature had a robust predictive performance.

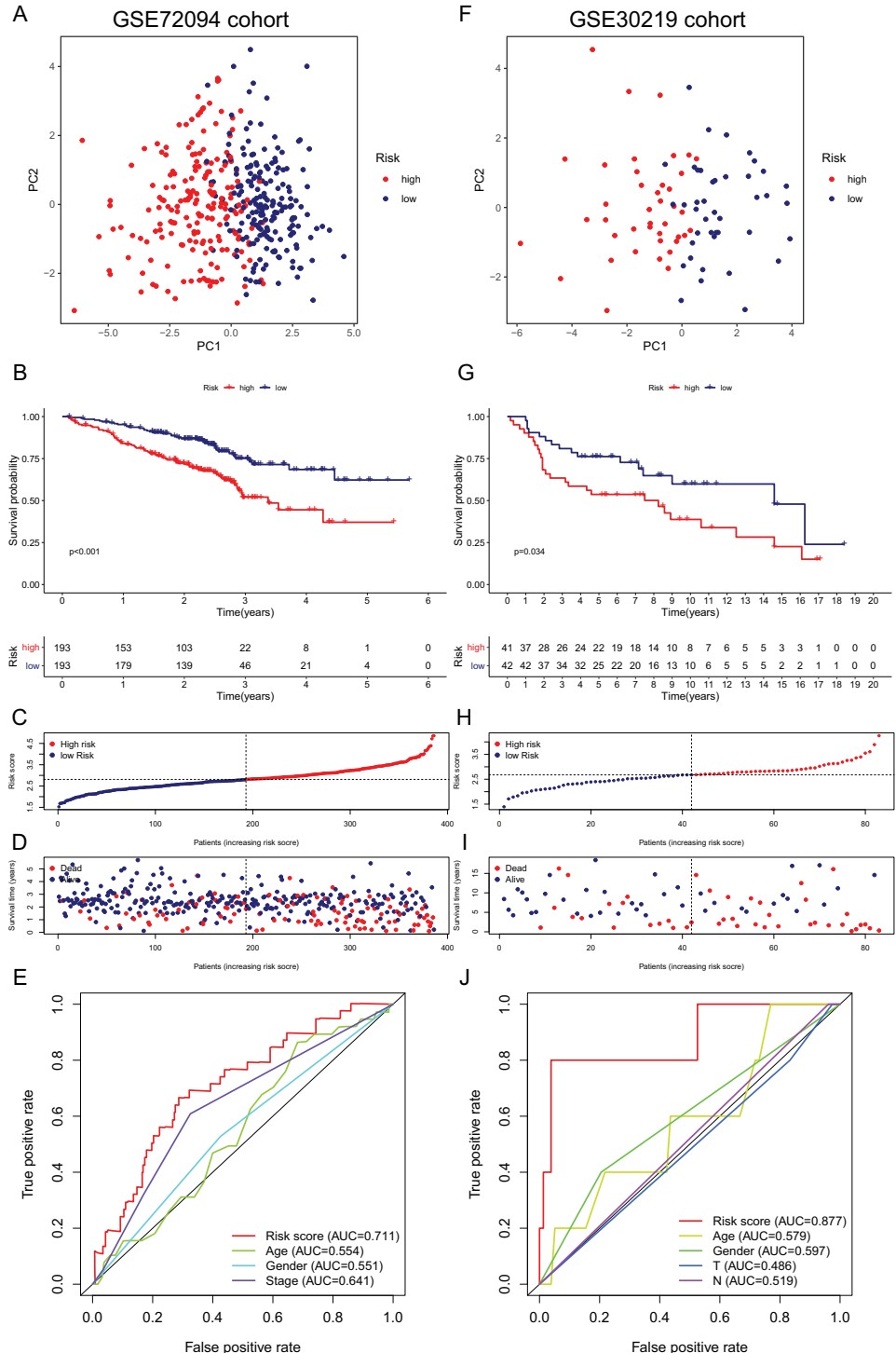

**Figure 5 Validation of the 15-gene ferroptosis signature in the GSE72094 cohort (A–E) and GSE30219 cohort (F–J).** (A, F) PCA plots of patients in high-risk group (red) and low-risk group (navy blue). (B, G) Kaplan–Meier curves of patients in high-risk group (red) and low-risk group (navy blue). (C, H) The distribution of risk score of each patient. (D, I) The distribution of OS, survival status and risk score of each patient. (E, J) Time- dependent ROC curves of risk score and clinical characteristics. PCA, principal component analysis; OS, overall survival; AUC, area under curve; ROC, receiver operating characteristic.

**Table 1 Univariate and multivariate analysis of OS in the TCGA, GSE72094 and GSE30219 cohorts.**

| Variables | Univariate analysis | | | Multivariate analysis | | |
|---|---|---|---|---|---|---|
| | HR | 95% CI | *P* value | HR | 95% CI | *P* value |
| TCGA cohort | | | | | | |
| Age | 0.997 | [0.978–1.015] | 0.718 | | | |
| Gender | 1.000 | [0.694–1.441] | 1.000 | | | |
| Stage | 1.648 | [1.396–1.946] | <0.001 | 1.591 | [1.064–2.380] | 0.024 |
| T | 1.600 | [1.285–1.994] | <0.001 | 1.064 | [0.830–1.364] | 0.626 |
| N | 2.765 | [1.911–4.001] | <0.001 | 1.316 | [0.723–2.397] | 0.369 |
| M | 1.748 | [0.959–3.187] | 0.068 | | | |
| Risk score | 3.670 | [2.521–5.341] | <0.001 | 2.918 | [1.972–4.318] | <0.001 |
| GSE72094 cohort | | | | | | |
| Age | 1.010 | [0.990–1.030] | 0.344 | | | |
| Gender | 1.501 | [1.027–2.194] | 0.036 | 1.547 | [1.050–2.280] | 0.027 |
| Stage | 1.628 | [1.358–1.951] | <0.001 | 1.589 | [1.351–1.920] | <0.001 |
| Risk score | 2.607 | [1.918–3.544] | <0.001 | 2.135 | [1.722–3.208] | <0.001 |
| GSE30219 cohort | | | | | | |
| Age | 1.031 | [0.996–1.067] | 0.088 | | | |
| Gender | 1.114 | [0.516–2.403] | 0.784 | | | |
| T | 1.463 | [0.856–2.499] | 0.164 | | | |
| N | 1.247 | [0.299–5.202] | 0.762 | | | |
| Risk score | 2.881 | [1.516–5.474] | 0.001 | 6.751 | [2.400–18.785] | <0.001 |

**Note:**
Abbreviations: TCGA, The Cancer Genome Atlas; HR, hazard ratio; CI, confidence interval.

## Independent prognostic value of the ferroptosis-related gene signature

To further assess the independent prognostic value of the 15-gene ferroptosis signature, we performed univariate and multivariate Cox regression analyses on all characteristics, including age, sex and TNM stage, and the risk score model based on the ferroptosis signature in both the derivation cohort and two validation cohorts. The results are presented in Table 1. In the TCGA cohort, TNM stage and risk score model were identified to be independent prognostic factors for OS. In the GSE72094 cohort, sex, TNM stage and risk score model were independent prognostic factors for OS. The risk score model was the only independent prognostic factor for OS in the GSE30219 cohort.

Given that the risk score model was not the only independent prognostic factor in the TCGA and GSE72094 cohorts, we further performed subgroup survival analysis to verify whether the 15-gene ferroptosis signature could be independent of other risk factors to predict prognosis. In the TCGA cohort, patients in stage I–II and stage III–IV were stratified into high-risk and low-risk groups by the median risk score. K–M curves showed that the OS of the high-risk group was significantly worse than that of the low-risk group regardless of whether patients were stage I-II or stage III-IV, indicating that the 15-gene ferroptosis signature could predict OS independent of TNM stage (Figs. 6A, 6B).

Similarly, in the GSE72094 cohort, the OS of the high-risk group was significantly worse than that of the low-risk group regardless of whether the patient was male or female, indicating that the signature could predict OS independent of sex (Figs. 6C, 6D). For the TNM stage subgroup, the OS of patients in the high-risk group was significantly shorter than that of patients in the low-risk group except for patients in stage III and stage IV, which was likely due to the small sample size of patients in these groups (Figs. 6E, 6F). Overall, the 15-gene ferroptosis signature could predict OS independent of other clinical characteristics.

## Construction and assessment of the predictive nomogram

Based on the independent risk factors identified in the TCGA cohort, we constructed a nomogram model to predict the OS probabilities of patients at 1, 2 and 3 years for risk assessment and earlier intervention to improve patient survival time (as shown in Fig. 7A). As depicted in Fig. 7B, a significantly poorer prognosis was observed in the high-risk group. The C index for the nomogram model to predict OS was 0.770 (95% CI [0.724–0.816]), indicating that the nomogram model had a robust predictive accuracy. We further evaluated the discrimination and calibration of the nomogram by time-dependent ROC curves and calibration plots for 1-year and 3-year OS. The 1-year and 3-year calibration plots indicated that the predicted survival was highly consistent with the actual survival in the TCGA cohort (Figs. 7D, 7E). As shown in Figs. 7H and 7I, the AUC values of the nomogram for predicting 1-year and 3-year OS were 0.812 and 0.757, respectively. All the AUC values of the nomogram were superior to those of other independent risk factors, indicating that the nomogram had a better predictive performance.

The nomogram was further validated in the GSE72094 cohort, while TNM stage information was not given in the GSE30219 cohort. The results showed that the OS of patients in the high-risk group was significantly shorter than that of patients in the low-risk group (Fig. 7C). The C index of the nomogram model was 0.708 (95% CI [0.660–0.756]). The calibration plots indicated great consistency between the predicted survival rate and the real observation (Figs. 7F, 7G). The AUC values of the nomogram for predicting 1-year and 3-year OS were 0.735 and 0.758, respectively, which were also superior to those of all other independent risk factors (Figs. 7J, 7K).

## Functional analysis of the 15-gene ferroptosis signature

GO and KEGG enrichment analyses were performed to elucidate the possible biological functions and pathways involved in the 15-gene ferroptosis signature. The results of GO enrichment analysis indicated that the DEGs between the high-risk group and the low-risk group were mainly enriched in pathways of the cell cycle and immune response, such as chromosome segregation, mitotic nuclear division, mitotic sister chromatid segregation, antimicrobial humoral response and humoral immune response (Fig. 8A and Table S7). As shown in Fig. 8B and Table 2, the KEGG enrichment analysis revealed that the top five pathways enriched in the high-risk group were the cell cycle, ubiquitin-mediated proteolysis, oocyte meiosis, homologous recombination and p53 signaling

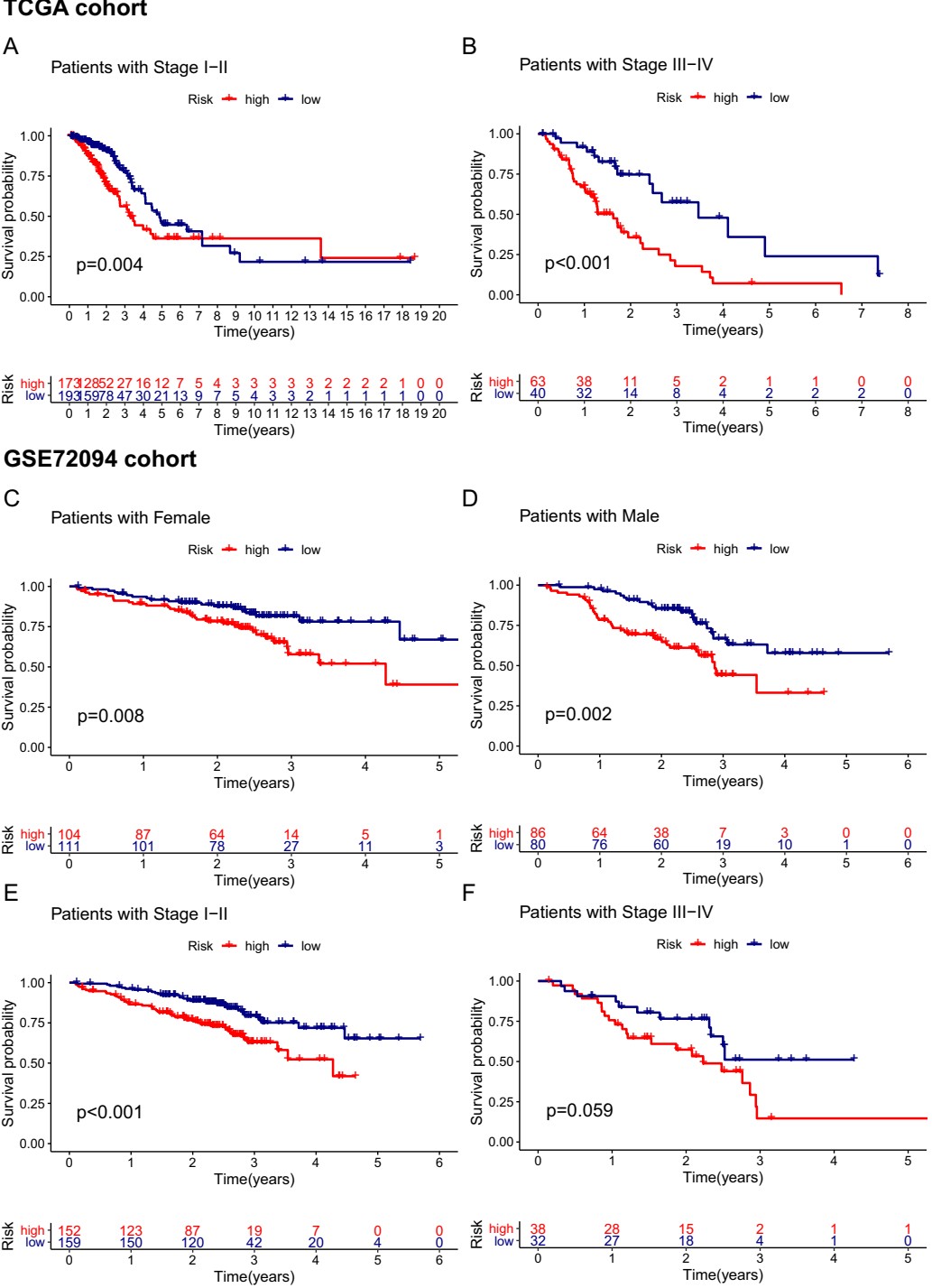

**Figure 6 Stratification analyses of the TCGA cohort (A–B) and GSE72094 cohort (C–F).** (A, B) Kaplan–Meier curves of patients in high-risk group (red) and low-risk group (navy blue) stratified via stage I-II and stage III-IV in the TCGA cohort, respectively. (C, D) Kaplan–Meier curves of patients in high-risk group (red) and low-risk group (navy blue) stratified via female and male in the GSE72094 cohort, respectively. (E, F) Kaplan–Meier curves of patients in high-risk group (red) and low-risk group (navy blue) stratified via stage I-II and stage III-IV in the GSE72094 cohort. TCGA, The Cancer Genome Atlas.

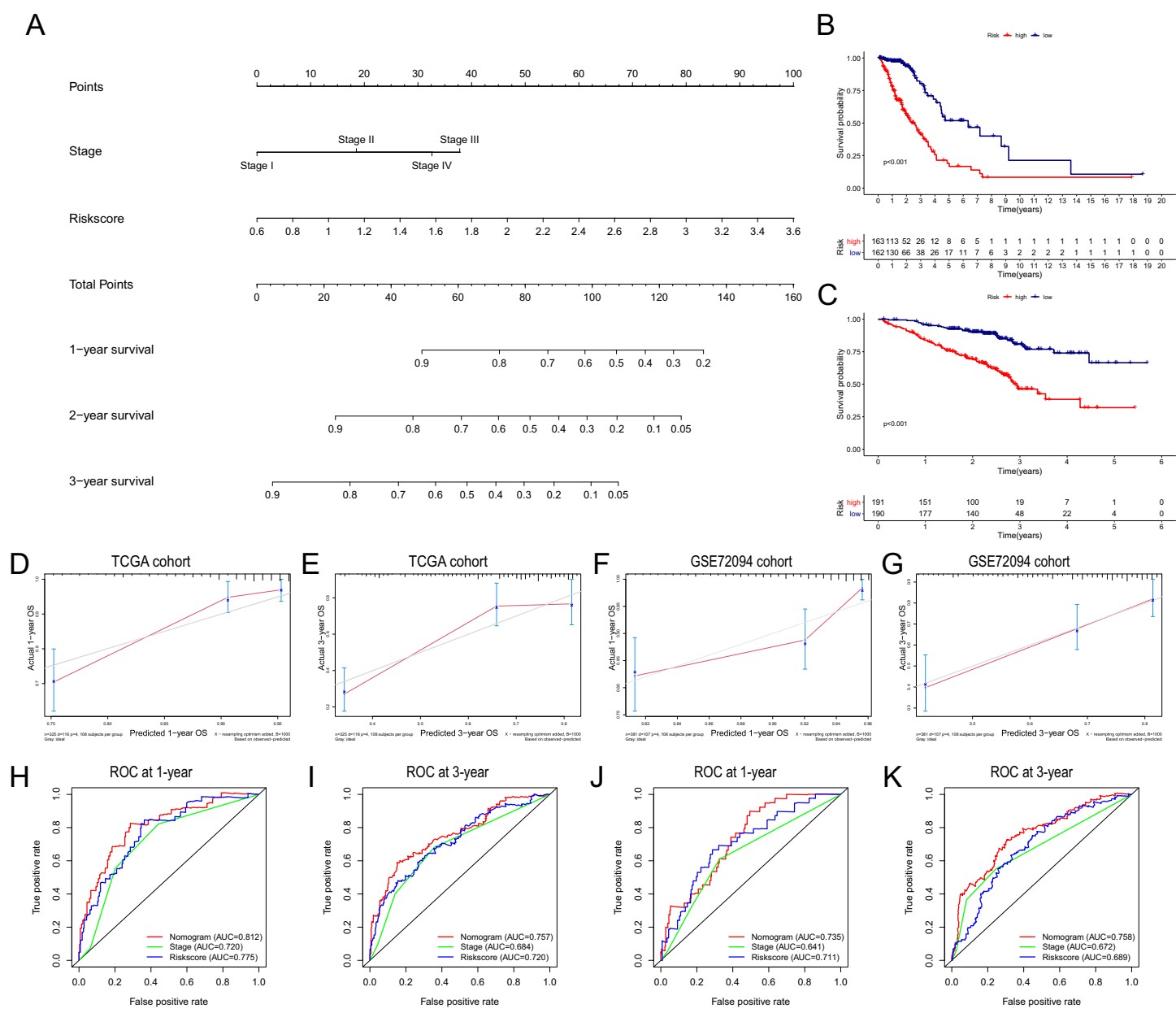

**Figure 7 Construction and validation of the predictive nomogram.** (A) The nomogram for OS prediction at 1, 2 and 3 years was constructed in the TCGA cohort. (B) Kaplan–Meier curve of patients in high-risk group (red) and low-risk group (navy blue) in the TCGA cohort. (C) 2xKaplan–Meier curve of patients in high-risk group (red) and low-risk group (navy blue) in the GSE72094 cohort. (D, E) Calibration plots of nomogram for OS prediction at 1 and 3 years in the TCGA cohort, respectively. (F, G) Calibration plots of nomogram for OS prediction at 1 and 3 years in the GSE72094 cohort, respectively. (H, I) Time-dependent ROC curves to evaluate the predictive performance of nomogram at 1 and 3 years in the TCGA cohort, respectively. (J, K) Time-dependent ROC curves to evaluate the predictive performance of nomogram at 1 and 3 years in the GSE72094 cohort, respectively. In the calibration plot, the closer the red line (fitting line) and gray line (ideal line) are, the higher predictive accuracy of the model is. TCGA, The Cancer Genome Atlas; OS, overall survival; AUC, area under curve; ROC, receiver operating characteristic.

pathways. The top five pathways enriched in the low-risk group were the arachidonic acid metabolism, primary bile acid biosynthesis, alpha linolenic acid metabolism, asthma, and intestinal immune network for IgA production pathways.

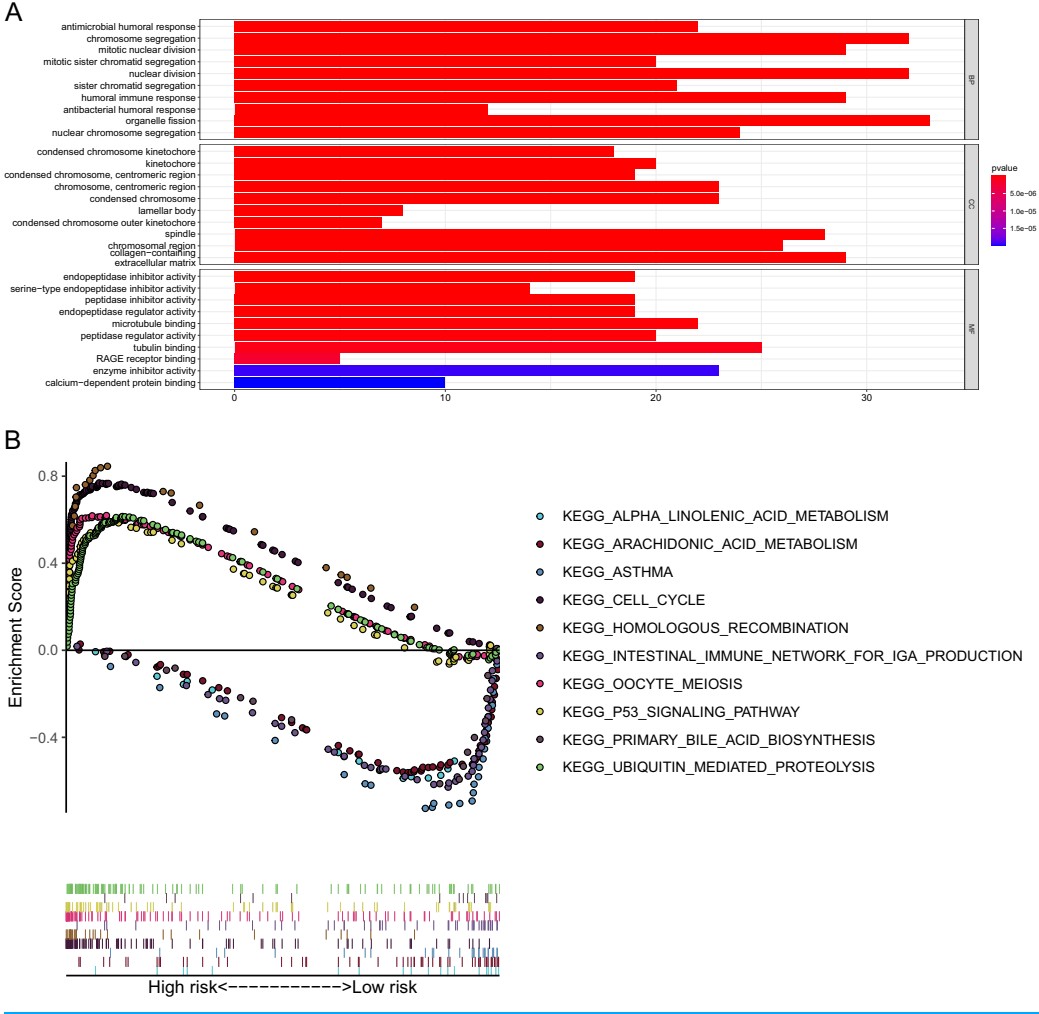

**Figure 8 GO and KEGG pathway enrichment analyses.** (A) Representative GO pathways enriched in the DEGs between the high- and low-risk groups. (B) The top five enriched KEGG pathways identified by GSEA in the high- and low-risk groups. |log2FC| ≥ 1 and FDR < 0.05 were set as the criteria to screen DEGs between the high- and low-risk groups. The pathways with a P value less than 0.05 were considered to be significantly enriched. DEGs, differentially expressed genes; GO, Gene Ontology; BP, Biological Process; CC, Cellular Component; MF, Molecular Function; KEGG, Kyoto Encyclopedia of Genes and Genomes; GSEA, Gene Set Enrichment Analysis. 

## Correlation with immune status

Given that pathways of the immune response were significantly enriched in the 15-gene ferroptosis signature, we further sought to investigate the correlation between immune status and risk score of the ferroptosis signature. The enrichment results are presented in Tables S8–10. Then, we analyzed the difference in the enrichment score of immune cells and immune functions between the high-risk and low-risk groups. By comparing the enrichment results of the three cohorts, we found that enrichment scores of activated dendritic cells (aDCs), dendritic cells (DCs), immature dendritic cells (iDCs), mast cells and neutrophils were significantly lower in the high-risk group than in the low-risk group (Figs. 9A, 9C, 9E). Regarding immune-related functions, higher enrichment score of

**Table 2 Top five enriched KEGG pathways of the high-risk and low-risk groups analysed by GSEA.**

| Enriched pathways | Size | Es | NES | NOM p-value | FDR q-value |
|---|---|---|---|---|---|
| High-risk group | | | | | |
| KEGG_CELL_CYCLE | 125 | 0.769 | 2.495 | <0.001 | <0.001 |
| KEGG_UBIQUITIN_MEDIATED_PROTEOLY-SIS | 135 | 0.615 | 2.458 | <0.001 | <0.001 |
| KEGG_OOCYTE_MEIOSIS | 113 | 0.619 | 2.342 | <0.001 | <0.001 |
| KEGG_HOMOLOGOUS_RECOMBINATION | 28 | 0.845 | 2.291 | <0.001 | <0.001 |
| KEGG_P53_SIGNALING_PATHWAY | 68 | 0.589 | 2.270 | <0.001 | <0.001 |
| Low-risk group | | | | | |
| KEGG_ARACHIDONIC_ACID_METABOLISM | 58 | 0.571 | 1.988 | 0.002 | 0.103 |
| KEGG_PRIMARY_BILE_ACID_BIOSYNTHE-SIS | 16 | 0.661 | 1.883 | <0.001 | 0.144 |
| KEGG_ALPHA_LINOLENIC_ACID_METABO-LISM | 19 | 0.621 | 1.823 | 0.006 | 0.156 |
| KEGG_ASTHMA | 28 | 0.745 | 1.820 | 0.017 | 0.119 |
| KEGG_INTESTINAL_IMMUNE_NETWORK_ FOR_IGA_PRODUCTION | 46 | 0.650 | 1.740 | 0.040 | 0.177 |

Note:
Abbreviations: ES, enrichment score; NES, normalized enrichment score; NOM p-value, nominal p value; FDR q-valve, false discovery rate.

antigen-presenting cell (APC) inhibition was found in the high-risk groups of all three cohorts, while a higher enrichment score of the type II IFN (IFN-γ) response was found in the low-risk group (Figs. 9B, 9D, 9F). All these results indicated a close correlation between risk score of the ferroptosis signature and immune cells as well as immune functions.

# DISCUSSION

Ferroptosis is a novel form of programmed cell death characterized by the excessive accumulation of intracellular iron and an increase in reactive oxygen species (ROS) (*Dixon et al., 2012*). Disturbances in iron hemostasis lead to excessive intracellular iron accumulation and may induce ferroptosis (*Bogdan et al., 2016*). In recent years, this unique pattern of cell death has been the focus of a large number of studies, and it is well characterized as a promising therapeutic alternative for various cancer types (*Hassannia, Vandenabeele & Vanden Berghe, 2019*). However, there are still limited studies regarding the specific role of ferroptosis in LUAD as well as its potential mechanism and pathways. The rapid development of RNA sequencing and microarrays and large-scale public databases provide an opportunity to obtain a better understanding of these ferroptosis-related genes and to construct a reliable ferroptosis-based prognostic signature.

In the present study, we systematically analyzed the expression profile of ferroptosis-related genes in LUAD, and we found that the majority of ferroptosis-related genes (77.6%, 97/125) were differentially expressed between LUAD tumor tissues and normal tissues. Univariate Cox regression analysis showed that 22 genes were associated with OS, and LASSO Cox regression finally identified 15 genes (AGER, CISD1, DPP4, EGLN1, FANCD2, GLS2, ISCU, ITGA6, ITGB4, KRAS, NEDD4, PEBP1, SLC11A2, TFAP2A and VDAC1) to construct the ferroptosis-related gene signature. Their modulatory effects on ferroptosis are summarized in Table 3. Among them, CISD1, DPP4, EGLN1, FANCD2,

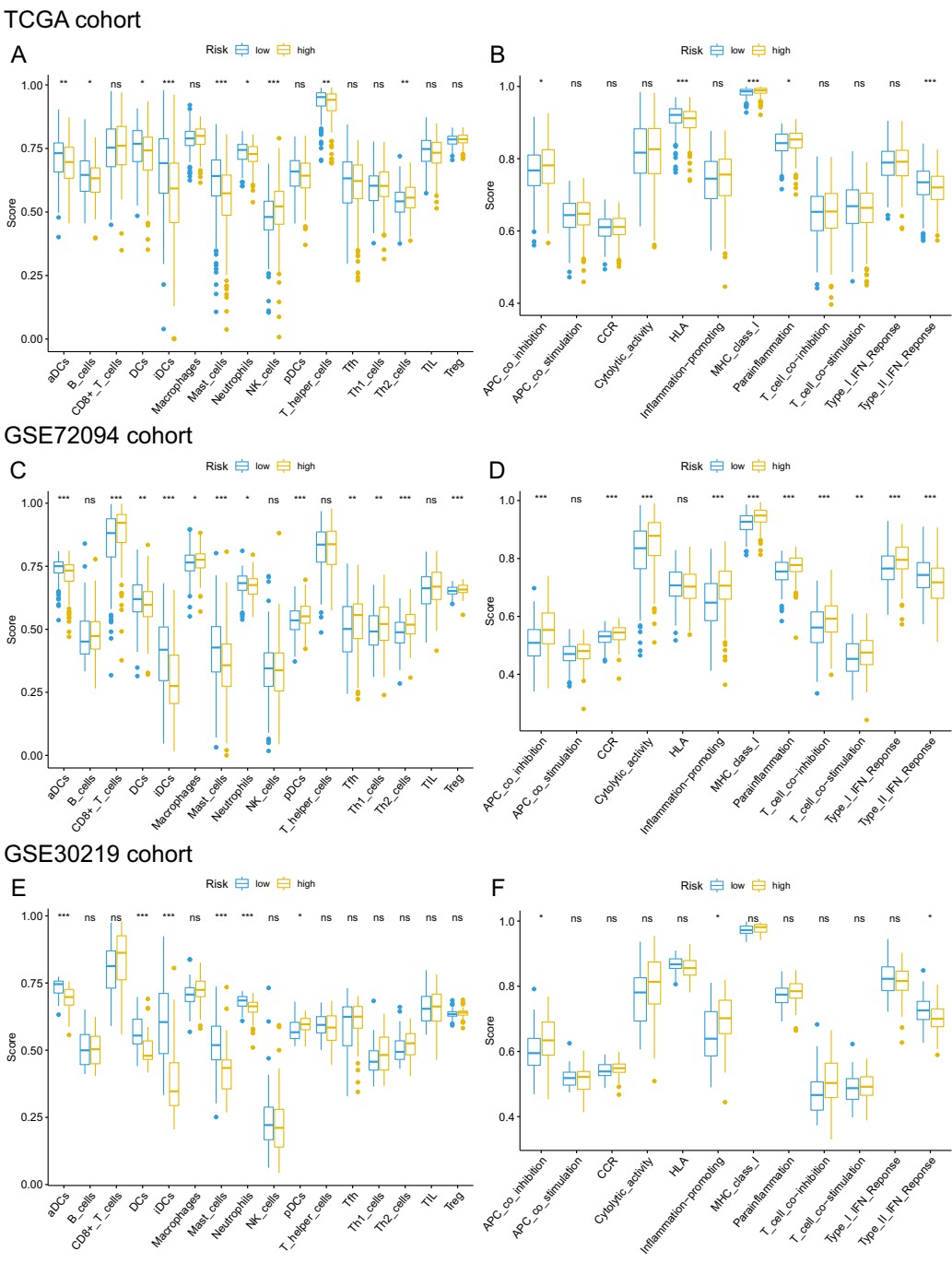

**Figure 9 Comparison of ssGSEA scores of different risk groups in the TCGA cohort (A, B), GSE72094 cohort (C, D) and GSE30219 cohort (E, F).** (A, C, E) Boxplots to display the enrichment scores of 16 immune cells of different risk groups in the TCGA cohort, GSE72094 cohort and GSE30219 cohort, respectively. (B, D, F) Boxplots to display the enrichment scores of 12 immune functions of different risk groups in the TCGA cohort and GSE72094 cohort and GSE30219 cohort, respectively. Adjusted *P* values were showed as: ns, not significant; *, *P* < 0.05; **, *P* < 0.01; ***, *P* < 0.001. ssGSEA, single-sample gene set enrichment analysis; TCGA, The Cancer Genome Atlas; DC, Dendritic cells; APC, Antigen-presenting cells; Type II INF, IFN-γ.

ISCU, ITGA6, ITGB4, KRAS and NEDD4 negatively regulated ferroptosis, while AGER, GLS2, PEBP1, SLC11A2 and VDAC1 positively regulated ferroptosis. The depletion of AGER inhibited autophagy-dependent ferroptosis (*Wen et al., 2019*). Low CISD1 expression could contribute to the ferroptosis of hepatocellular cancer cells by iron-mediated intramitochondrial lipid peroxidation (*Yuan et al., 2016*). The accumulation of DPP4 could protect cells from ferroptosis by inhibiting lipid peroxidation in human colorectal cancer (*Xie et al., 2017*). A previous study indicated that EGLN1 could act as an oncogene by inducing LSH expression, which inhibits ferroptosis in lung cancer (*Jiang et al., 2017*). FANCD2 was found to act as a ferroptosis suppressor by decreasing lipid peroxidation (*Song et al., 2016*). Upregulation of GLS2 exhibited antitumor effects in gastric cancer by promoting ferroptosis (*Niu et al., 2019*). ISCU, a mitochondrial protein, could regulate iron metabolism and increase the expression of GSH, thus significantly alleviating ferroptosis (*Du et al., 2019*). Integrin subunit alpha 6 (ITGA6) and integrin subunit beta 4 (ITGB4) belong to the integrin family, which mainly functions in cell adhesion. Overexpression of ITGA6 and ITGB4 attenuated the ferroptosis induced by erastin, while knockout of ITGA6 and ITGB4 promoted ferroptosis in breast cancer cells (*Brown et al., 2017*). Activated mutant KRAS could promote the expression of SCL7A11, thus inhibiting ferroptosis in lung adenocarcinoma (*Hu et al., 2020*). The depletion of NEDD4 was found to promote the ferroptosis induced by erastin by limiting the degradation of VDAC2/3 in melanoma (*Yang et al., 2020*). PEBP1, a scaffold protein kinase cascade inhibitor, was reported to promote ferroptosis by enabling lipoxygenase (*Wenzel et al., 2017*). SLC11A2, also called DMT1, is a major iron transporter and contributes to iron uptake in most cell types, and its upregulation could trigger ferroptosis in acute myocardial infarction mice (*Song et al., 2020*). TFAP2A was reported to negatively modulate ferroptosis by activating the NRF2 signaling pathway (*Huang et al., 2020*). Inhibition of VDAC1 could significantly alleviate ferroptosis and improve cell survival by decreasing the ROS level in mitochondria (*Nagakannan et al., 2019*). Subsequent K–M survival analyses confirmed the prognostic value of the 15 ferroptosis-related genes, indicating their potential role in the initiation and progression of LUAD. Unsurprisingly, the risk scores of the 15-gene ferroptosis signature were demonstrated to be significantly associated with the OS of LUAD patients in both the TCGA cohort and two external validation cohorts. The signature was independent of other clinical characteristics in all three cohorts. In addition, the nomogram integrating the independent risk factors, including the risk score model, exhibited high predictive value and may help clinicians make optimal clinical decisions to improve the OS rate of LUAD patients. These findings indicated the important role of ferroptosis in the progression of LUAD and the possibility of the ferroptosis signature as a biomarker for OS.

KEGG pathway analysis showed that the high-risk group was mainly enriched in pathways closely associated with tumorigenesis, such as the cell cycle, oocyte meiosis and homologous recombination. GO analysis of the DEGs between the high-risk group and

**Table 3 Modulatory effect of the signature genes on ferroptosis.**

| Gene symbol | Protein | Modulatory effect on ferroptosis | References |
|---|---|---|---|
| AGER | Advanced glycosylation end product-specific receptor | Depletion attenuates the autophagy-dependent ferroptosis | Wen et al. (2019) |
| CISD1 | CDGSH iron-sulfur domain-containing protein 1 | Down-expression contributes to ferroptosis by iron-mediated intramitochondrial lipid peroxidation | Yuan et al. (2016) |
| DPP4 | Dipeptidyl peptidase 4 | Overexpression protects cell from ferroptosis by inhibiting lipid peroxidation | Xie et al. (2017) |
| EGLN1 | Egl nine homolog 1 | Overexpression inhibits ferroptosis by the induction of LSH | Jiang et al. (2017) |
| FANCD2 | Fanconi anemia group D2 protein | Knockout promotes ferroptosis though increasing iron accumulation and lipid peroxidation | Song et al. (2016) |
| GLS2 | Glutaminase liver isoform | Upregulation promotes ferroptosis in gastric cancer | Niu et al. (2019) |
| ISCU | Iron-sulfur cluster assembly enzyme ISCU | Overexpression attenuates ferroptosis by increasing the level of GSH | Du et al. (2019) |
| ITGA6 | Integrin alpha-6 | Overexpression inhibits ferroptosis by suppressing the expression of ACSL4 | Brown et al. (2017) |
| ITGB4 | Integrin beta-4 | Overexpression inhibits ferroptosis by suppressing the expression of ACSL4 | Brown et al. (2017) |
| KRAS | GTPase KRas | Inhibits ferroptosis by upregulating the expression of SLC7A11 | Hu et al. (2020) |
| NEDD4 | E3 ubiquitin-protein ligase NEDD4 | Upregulation inhibits ferroptosis induced by erastin through the degradation of VDAC2/3 | Yang et al. (2020) |
| PEBP1 | Phosphatidylethanolamine-binding protein 1 | Overexpression increases sensitivity to ferroptosis in HAEC and HT22 cells | Wenzel et al. (2017) |
| SLC11A2 | Natural resistance-associated macrophage protein 2 | Upregulation contributes to ferroptosis by increasing iron uptake | Song et al. (2020) |
| TFAP2A | Transcription factor AP-2-alpha | Negatively regulates ferroptosis by increasing the expression of NRF2 | Huang et al. (2020) |
| VDAC1 | Voltage-dependent anion-selective channel protein 1 | Inhibition alleviates ferroptosis by decreasing the ROS level in mitochondria | Nagakannan et al. (2019) |

low-risk group showed that the biological functions of the DEGs were mainly enriched in the regulation of the cell cycle and immune response, indicating a link between ferroptosis and antitumor immunity. Cancer growth and metastasis are closely related to interactions with the immune system (Matsushita et al., 2012). When investigating the correlation with immune status, we interestingly found that enrichment scores of aDCs, DCs, iDCs, mast cells and neutrophils were significantly lower in the high-risk group. Among these differentially enriched immune cells, iDCs specialize in antigen capture, and DCs are professional APCs (Lin et al., 2019; Tiberio et al., 2018). Both aDCs and iDCs play crucial roles in the process of cytotoxic T cell activation and in the regulation of the immune response to cancer cells (Durai & Murphy, 2016). Moreover, neutrophils infiltrating tumor tissues, termed tumor-associated neutrophils (TANs), also play a role in antitumor immunity (Lecot et al., 2019). It was confirmed that TANs could release cytotoxic substances such as ROX, thus directly inducing tumor cell apoptosis (Takeshima et al., 2016). The role of mast cells in the progression of tumors is controversial. For instance, mast cells could play a protumorigenic role by releasing angiogenic and

lymphangiogenic factors, which promote angiogenesis and lymphogenesis (*Detoraki et al., 2009*). In contrast, mast cells could secrete tumor necrosis factor α (TNF-α) and directly mediate tumor cell cytotoxicity, thus playing an antitumorigenic role (*Varricchi et al., 2017*). In lung cancer, Bao et al. and Welsh et al. reported that the low abundance of mast cell in cancer specimens was correlated with worse OS (*Bao et al., 2020*; *Welsh et al., 2005*). When we analyzed the difference in immune functions, a stronger inhibition of APCs was found in the high-risk groups of both the derivation cohort and two validation cohorts. The synergistic effect of reduced DC and stronger inhibition of APCs in the high-risk group may contribute to the great suppression of the tumor antigen presentation process. Additionally, we also found that enrichment score of type II IFN was significantly lower in the high-risk group. IFN-γ produced by APCs, T cells, B cells, NK cells and NKT cells is the only member of the type II INF family (*Castro et al., 2018*). It is well known that IFN-γ plays a pivotal role in cancer immune surveillance, stimulating antitumor immunity and facilitating the recognition and elimination of cancer cells (*Benci et al., 2019*; *Shankaran et al., 2001*; *Street, Cretney & Smyth, 2001*). All these findings suggested that ferroptosis may be involved in antitumor immune response. However, more experimental investigation is warranted to confirm these findings and reveal the underlying mechanism.

## CONCLUSIONS

The 15-gene ferroptosis signature identified in this study could be a potential biomarker for prognosis prediction in LUAD. Targeting ferroptosis might be a promising therapeutic alternative for LUAD.

## ACKNOWLEDGEMENTS

The authors are truly thankful to the TCGA and GEO databases for the availability of the data.

### Funding

This work was supported by the National Natural Science Foundation of China (No. 82072594), the Hunan Provincial Key Area R&D Programmes (No. 2019SK2253) and the Fundamental Research Funds for the Central Universities of Central South University (No. 2021zzts0385). The funders had no role in study design, data collection and analysis, decision to publish, or preparation of the manuscript.

### Grant Disclosures

The following grant information was disclosed by the authors:
National Natural Science Foundation of China: 82072594.
Hunan Provincial Key Area R&D Programmes: 2019SK2253.
Central Universities of Central South University: 2021zzts0385.

## Competing Interests

The authors declare that they have no competing interests.

## Author Contributions

- Guangxu Tu conceived and designed the experiments, performed the experiments, analyzed the data, prepared figures and/or tables, authored or reviewed drafts of the paper, and approved the final draft.
- Weilin Peng performed the experiments, analyzed the data, prepared figures and/or tables, and approved the final draft.
- Qidong Cai performed the experiments, prepared figures and/or tables, and approved the final draft.
- Zhenyu Zhao performed the experiments, prepared figures and/or tables, and approved the final draft.
- Xiong Peng performed the experiments, prepared figures and/or tables, and approved the final draft.
- Boxue He performed the experiments, prepared figures and/or tables, and approved the final draft.
- Pengfei Zhang performed the experiments, prepared figures and/or tables, and approved the final draft.
- Shuai Shi performed the experiments, prepared figures and/or tables, and approved the final draft.
- Yongguang Tao conceived and designed the experiments, authored or reviewed drafts of the paper, and approved the final draft.
- Xiang Wang conceived and designed the experiments, authored or reviewed drafts of the paper, and approved the final draft.

## Data Availability

The data is available at the TCGA database (https://portal.gdc.cancer.gov/) and NCBI GEO: GSE72094 and GSE30219.

## Supplemental Information

Supplemental information for this article can be found online at http://dx.doi.org/10.7717/peerj.11687#supplemental-information.

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
