# Peer review of "Construction and validation of a 15-gene ferroptosis signature in lung adenocarcinoma"

_PeerJ, doi:10.7717/peerj.11687_

## Round 0.1 · original submission · Major Revisions

The reviewers raised significant concerns regarding the experimental design and the validity of the findings. All these major points should be addressed before this manuscript can be accepted for publication.

Reviewer 1 ·

Basic reporting

A few minor points mentioned in the 'General comments for the author'.

Experimental design

A few minor points mentioned in the 'General comments for the author'.

Validity of the findings

This part is robust!

Additional comments

In this study, Tu et al. established a 15-genes ferroptosis signature. Ferroptosis is a relatively new cell death phenomenon that is dependent on iron and lip peroxides. From the TCGA patients database and a validation dataset, they concluded significantly shorter overall survival of the lung adenocarcinoma patients in the high-risk gene signature group based on the expression of this signature. These patients encountered a more potent immune suppression, especially in the antigen presentation process and a higher level of PD-L1 expression. This could serve as a biomarker for predicting the response to immunotherapy in lung adenocarcinoma. It is a promising study for a better-personalized treatment opportunity in lung adenocarcinoma patients. Though the in-vitro and in-vivo experiments need to be performed for validation, they are out of the scope of this in-silico study paper and are destined for future research. I would suggest the following recommendations to the authors in order to make this manuscript better:

1. A couple of lines about ferroptosis should be mentioned in the abstract as well. The authors described it in the introduction, but it is relatively a less studied phenomenon, so a brief description in the abstract would attract the readers.
2. If a new 'R' script is written for this manuscript, please consider it depositing in the public database like Github and give its reference in the manuscript.
3. As a general note, there is no space between the first brace of the references and the preceding word. A space should be incorporated. e.g. in line 51...cancer(Wei et al. 2018) should be cancer (Wei et al. 2018).
4. In line 107, elaborate on the criteria for discarding the unrelated genes.
5. Please replace volcano with volcano plots wherever mentioned e.g. in line 126.
6. Usually, in these types of studies, more than one validation cohorts are used. Here only GSE72094 as the external validation cohort is used. Why specifically this cohort, and why only one being used?
7. Figure legends are not very informative. Please expand them and add more information to make them self-explanatory.
8. It seems like the legends for the supplementary figures are missing. Could authors double-check it?
9. Fig. 1B is not very informative due to its small size. I would recommend the authors to keep it as it is in this figure as a representation and included the full-page zoomed version in the supplementary so that the names of differentially expressed genes are visible.
10. For the volcano plot in Fig. 1C, label the most significant top hits (4 to 7).
11. For the survival plots in Fig. 3 and elsewhere, mention the number of patients.
12. Please include a good discussion about the enriched common mutations based on Fig. 7. It is missing now!
13. From Fig. 5, some immune-related cells/functions are enriched in high-risk groups and some in low-risk groups. What biological significance could be drawn from this? This should be discussed.
14. A conclusion figure (graphical abstract) will be very useful for the readers.
15. In the discussion, from line 303 onwards about the description of the 15 genes, it would be better and more useful if the authors could present this description in a table format, including the references.
16. In line 53, ‘in’ is missing from Recent years.
17. Line 79- typo, a proximately.
18. Line 105- typo, iron iron
19. Reference for line 267 missing.
20. Line 286 is incomplete.

·

Basic reporting

Tu et al, report a ferroptosis -related signature in patients with lung adenocarcinoma. To begin with, the authors state an inadequate claim starting from the title. The response to immunotherapy is not supported at all from their data.

In the introduction section, the authors do not include important and highly impactful reports regarding the role of the SLC7A11 in lung cancer progression, stating that not many reports are available (line 72) (The Journal of clinical investigation, 130(4)., Oncogene, 37(36), pp.5007-5019., Redox biology, 38, p.101801., Cancer discovery, 9(12), pp.1673-1685.)

In several areas of the manuscript (line 51, 82, 263-269) the writing needs significant editing to improve comprehension. In many cases, the references used by the authors do not exist, displayed, properly interpreted (Tang et al., 2020, Huang et al 2019). IN other cases, teh authors use vague expressions to justify their findings and conceptual flow without citing literature accordingly (line 263).

The figures are not adequately prepared with poorly labelled graphs with limited flow between the different topics.

Experimental design

There are several concerns regarding the overall experimental design. The ‘validation’ cohort used in this study (GSE72094) contains mutated patients specifically, and many of them harbor KRas mutations. A recent report showed that inhibition of SLC7A11 in KRAS mutated lung adenocarcinoma causes synthetic lethality in these patients. Based on these, the conclusions of the analysis of this dataset is biased and may be affected by the dependency of the ferroptosis pathway in the biology of KRAS mutated lung adenocarcinoma. The authors are strongly advised to expand their ‘validation’ in several publicly available lung adenocarcinoma datasets in GEO.

In the methods section it is unclear how the authors performed the GSEA for the different cell types that they propose. They should also provide the GO IDs that they used for the latter analysis.

Validity of the findings

The proposed role of the gene signature in immunotherapy response is correlative. The fact that CD274 mRNA expression is differentially expressed between the groups does not mean that it affects the response to immunotherapy. The authors are advised to remove these claims.

Reviewer 3 ·

Basic reporting

In the present manuscript the authors used publicly available data bases in order to provide a novel 15-gene ferroptosis-related gene signature depicting the overall survival outcome as well as the possible response to immunotherapy in lung adenocarcinoma patients.


This is an interesting paper which highlights the role of ferroptosis in cancer progression and immunotherapy. Based on a 15 ferroptosis-related gene signature the patients of the two data bases used in this study were stratified into two risk groups with the ones in the high risk showing reduced OS accompanied by strong immunosuppression compared to the patients in the low risk group.


However, there are a few comments that should be addressed to further improve the manuscript.

Experimental design

1. The authors should consider providing the PPI network from the STRING database as well as the correlation network of the ferroptosis related genes identified in the present study.
2. Survival analyses of each of the 15 genes should be included in the second section of the results.
3. In the last section of the functional analysis, the DEGs between the two groups should be used for the Gene Ontology analysis in addition to the KEGG pathway.

Validity of the findings

The critical role of ferroptosis in cancer treatment is well established from numerous recent studies in different cancer types. However, this specific 15-gene signature is novel and this is the reason why the authors should provide additional information regarding the correlation network of these genes, the pathways they participate or their biological function, in order to give a possible explanation about the molecular mechanisms that might be responsible for the phenotypes presented in this manuscript.

---

## Round 0.2 · Minor Revisions

Please carefully address reviewer's concern regarding the validity of data interpretation.

Reviewer 1 ·

Basic reporting

-

Experimental design

-

Validity of the findings

-

Additional comments

I am satisfied with the revision of the authors. The manuscript can now be considered for publication!

·

Basic reporting

Tu et al., return a significantly revised and improved manuscript of their work on the ferroptosis signature in patients with lung adenocarcinoma.
Although the manuscript has been improved significantly, eliminating many mistakes and correlative claims, still there are some important points that should be revised.
The main concern is that in several sections of the manuscript, the effect that the authors trying to present is purely correlative.

Experimental design

The new proposed experiments are well designed. All the raw data are nicely organized and provided in the manuscript.
The authors have done a solid work on providing detailed and transparent supportive data. In the previous version, the experimental designed was poorly biased, a phenomenon much improved in this version of the manuscript.

My main concern is still focusing on the ssGSEA. I appreciate the fact that the new method is well explained and provided in Github.
Nevertheless, the interpretation of the data is inaccurate and does not support the clinical claim that the authors wish to state.

Validity of the findings

In line 283 the authors identify in an unbiased way, that the high expression of the ferroptosis signature correlates with immune response and immune processes.

Furthermore, in figure 9, the authors present the expression of several gene-signature that are markers of immune cell functions in the high and lo-w groups.

Nevertheless, in line 295 the authors interpret these data as different infiltration of the patient tumors by these immune cell lines. The authors have no data whatsoever to support the claim of immune infiltration. This is just a correlation, not a causation.

The fact that the ferroptosis signature is oncogenic in patients with lung adenocarcinoma, could clearly affect the clinical status of these patients, leading to the differential immune system function. Furthermore, the fact that the proposed signature is oncogenic, could also correlate with more aggressive therapeutic regiment, know to affect the immune cell population, function and distribution in these patients.
This is a very important point that is inaccurate and needs to be corrected.
The authors need to remove these claims for the discussion section as well.


The authors should only interpret these finding as they really are. A correlation of the ferroptosis signature with a proposed immune cellular function. Still a very important finding, that needs a crisp and precise description.

Additional comments

The manuscript is significantly improved and more impactful. If the authors edit some important data interpretation , I would support the publication fo this interesting concept.

Reviewer 3 ·

Basic reporting

The authors have addressed all my concerns and therefore I support the publication of this manuscript.

Experimental design

No comment

Validity of the findings

No comment

---

## Round 0.3 · accepted · Accept

Please note that PeerJ does not provide editing services before publication, so please carefully check your manuscript to be publication-ready.

·

Basic reporting

No comments.

Experimental design

No comments.

Validity of the findings

No comments.

Additional comments

The authors return a revised manuscript with a very careful and accurate interpretation of their data.
I recommend the publication of this interesting concept.
Congratulation to the authors.